**Data Availability Statement:** Due to national regulations in the Province of Quebec (Canada) and restrictions from the Centre de recherche du Centre

# Effects of intraoperative hemodynamic management on postoperative acute kidney injury in liver transplantation: An observational cohort study

**François Martin Carrier**[1,2,3]*, **Marie-Pierre Sylvestre**[4☯], **Luc Massicotte**[1], **Marc Bilodeau**[5], **Michaël Chassé**[2,3☯]

**1** Department of Anesthesiology, Centre Hospitalier de l'Université de Montréal (CHUM), Montréal, Quebec, Canada, **2** Department of Medicine–Intensive Care Division, Centre Hospitalier de l'Université de Montréal (CHUM), Montréal, Quebec, Canada, **3** Centre de Recherche du Centre Hospitalier de l'Université de Montréal (CRCHUM), Montréal, Quebec, Canada, **4** School of Public Health of University of Montreal, Montréal, Quebec, Canada, **5** Department of Medicine—Liver diseases division, Centre Hospitalier de l'Université de Montréal (CHUM), Montréal, Quebec, Canada

☯ These authors contributed equally to this work.

* francois.martin.carrier@umontreal.ca

## Abstract

### Background

Intraoperative restrictive fluid management strategies might improve postoperative outcomes in liver transplantation. Effects of vasopressors within any hemodynamic management strategy are unclear.

### Methods

We conducted an observational cohort study on adult liver transplant recipients between July 2008 and December 2017. We measured the effect of vasopressors infused at admission in the intensive care unit (ICU) and total intraoperative fluid balance. Our primary outcome was 48-hour acute kidney injury (AKI) and our secondary outcomes were 7-day AKI, need for postoperative renal replacement therapy (RRT), time to extubation in the ICU, time to ICU discharge and survival up to 1 year. We fitted models adjusted for confounders using generalized estimating equations or survival models using robust standard errors. We reported results with 95% confidence intervals.

### Results

We included 532 patients. Vasopressors use was not associated with 48-hour or 7-day AKI but modified the effects of fluid balance on RRT and mortality. A higher fluid balance was associated with a higher need for RRT (OR = 1.52 [1.15, 2.01], p<0.001 for interaction) and lower survival (HR = 1.71 [1.26, 2.34], p<0.01 for interaction) only among patients without vasopressors. In patients with vasopressors, higher doses of vasopressors were associated with a higher mortality (HR = 1.29 [1.13, 1.49] per 10 µg/min of norepinephrine).

hospitalier de l'Université de Montréal Review Ethic Board (REB), health medical data cannot be made available publicly. However, complete access to the research dataset is possible for research purpose after appropriate privacy agreements between research parties have been completed. Data access requests may be sent to the corresponding author (francois.martin.carrier@umontreal.ca), or directly to our REB (ethique.recherche.chum@ssss.gouv. qc.ca). Any further information may be provided by our REB head (marie.josee.bernardi.chum@ssss. gouv.qc.ca). The study reported in this manuscript is registered with the study number 19.113 and complied to all local regulations. The dataset is named "TOFLIQUIDE_net.RData" and is localized in our local research directories.

**Funding:** Dr Carrier (FMC) received a grant by the Fondation du CHUM ("Don d'organes et transplantation" program) to complete this work. Dr Chassé (MC) and Dr Sylvestre (MPS) are recipients of a Career Award Junior 1 from the Fonds de la Recherche du Québec – Santé. The funders had no role in study design, data collection and analysis, decision to publish, or preparation of the manuscript.

**Competing interests:** The authors have declared that no competing interests exist.

**Abbreviations:** ALF, Acute Liver Failure; AKI, Acute Kidney Injury; CI, Confidence Interval; CIT, Cold Ischemia Time; CVP, Central Venous Pressure; GEE, Generalized Estimating Equations; HR, Hazard Ratio; ICU, Intensive Care Unit; MELD, Model for End-stage Liver Disease; KDIGO, Kidney Diseases: Improving Global Outcomes; POR, Proportional Odds Ratio; OR, Odds Ratio; RRT, Renal Replacement Therapy.

## Conclusion

The presence of any vasopressor at the end of surgery was not associated with AKI or RRT. The use of vasopressors might modify the harmful association between fluid balance and other postoperative outcomes. The liberal use of vasopressors to implement a restrictive fluid management strategy deserves further investigation.

## Introduction

Liver transplantation is the second most performed solid organ transplantation in the world and is the only available treatment of end-stage liver disease and liver cancer [1, 2]. In the past decade, while liver transplantation postoperative survival have increased, postoperative complications increased as well [3–6]. Since human organs available for transplantation are rare, improving recipients' postoperative outcomes is an important objective.

Several perioperative events and factors are associated with complications after liver transplantation [5, 7, 8]. Among them, hypotension and vasopressor administration have both been associated with several harmful effects such as an increased incidence of acute kidney injury (AKI), graft failure and mortality [7–12]. We previously suggested that restrictive fluid management strategies might be associated with less postoperative complications in liver transplantation [13].

In an observational cohort study we recently published, we suggested that a higher fluid balance was associated with longer stay in the intensive care unit (ICU) and a lower survival in liver transplantation [14]. We did not include vasopressors as one of our exposures. However, any restrictive fluid management strategy is associated with an increased use of vasopressors [15, 16]. Since restricting fluid administration might be beneficial in liver transplant recipients [13, 17], but vasopressor administration seems to be associated with more complications in some observational studies [5, 7, 8], the specific role of vasopressor use alongside any fluid management strategy is not well-defined.

Two recent systematic reviews in patients undergoing a major surgery suggested that cardiac output-guided fluid management compared to any fixed fluid administration strategy reduce postoperative complications by 20–30% [18, 19]. In a third meta-analysis on intraoperative cardiac output-guided management clinical trials in major surgery, the greatest reduction of postoperative AKI came from a more liberal use of vasopressors and inotropes and not from any difference in fluid balance [20]. Patients undergoing liver transplantation were excluded from all these studies.

The primary objective of this study was to evaluate the association between intraoperative hemodynamic management and postoperative acute kidney injury (AKI) in adult liver transplant recipients. Our secondary objectives were to explore the same associations on other patients' postoperative outcomes.

## Methods

### Setting and participants

After approval by the *Centre de recherche du Centre hospitalier de l'Université de Montréal* Institutional Review Board, we conducted an observational cohort study at the Centre hospitalier de l'Université de Montréal [14]. We previously published some data from this cohort [14]. We thus gathered new variables from patients' charts and conducted new analyses to

address our new study objective. We extracted data between July 2017 and December 2019. We included all adult liver transplant recipients who had their surgery between July 2008 and December 2017. We excluded both patients who underwent two solid organ transplantations simultaneously (liver + kidney or liver + lung) and patients who were on renal replacement therapy (RRT) before their transplantation.

Organ donation, allocation and procurement was managed by our national organ procurement organization (Transplant Québec (www.transplantquebec.ca/en)). None of the transplant donors was from a vulnerable population and all donors or next of kin provided written informed consent that was freely given. Donation after circulatory death is only conducted after palliative care in our province. Donors were either previously registered in one of the national organ donor registries (*Régie de l'assurance maladie du Québec* or the *Chambre des notaires du Québec* organ donor registry (https://www.transplantquebec.ca/node/107)) and/or were consented by a recognized surrogate decision maker (see S1 Appendix, for donor consent form). Since the Province of Quebec has a universal public healthcare system, no financial reimbursement was provided to donors families.

## Exposures

Our hemodynamic management exposure was defined as the combination of vasopressors infused at ICU admission and the total intraoperative fluid balance. We used the presence of vasopressors at ICU admission for three reasons: all patients received vasopressors at some point during a liver transplantation, vasopressors at this time point might better reflect the whole fluid management strategy than vasopressor doses at any other time point (a liberal approach would be associated with less vasopressors at the end of surgery) and data at this time point was very well collected in our center (most of the time, intraoperative doses of vasopressors are not reported in patient charts). Fluid balance was defined as the total volume of fluid and all blood products received (including cellsaver output transfused) to which we subtracted the diuresis, the volume of drained ascites and the total volume of bleeding measured in the surgical suction canisters [14]. At our institution, the anesthesiology procedure included a phlebotomy before the surgery in patients with a hemoglobin concentration above 85 g/L and a normal renal function; this blood was transfused back to the patient in the reperfusion phase (see S2 Appendix) [21, 22]. We considered the performance of such a phlebotomy as an important component of the fluid management strategy: we included it in all analyses and used it to build a propensity score in some analyses.

Use of vasopressors was thus an exposure of interest and a potential effect modifier of the fluid balance effect on our postoperative outcomes (see S1 Fig in S2 Appendix). Because vasopressors at the end of surgery and fluid balance might also be markers of the severity of hemodynamic instability and of reperfusion syndrome, we first evaluated the aforementioned effect modification and then analyzed vasopressors as potential confounders if no effect modification was observed [15, 23–25]. We also converted all vasopressors to norepinephrine equivalent (in ug per minute) and combined together to create a single total vasopressor dose variable (that we used in subgroup and sensitivity analyses) that we used in some stratified analyses: vasopressin was converted from units per hour to ug per minute in a 1:500 ratio (0.02 units per minute of vasopressin = 10 ug per minute of norepinephrine) and phenylephrine was converted in a 13:1 ratio (100 ug per minute of phenylephrine = 8 ug per minute of norepinephrine) [26, 27].

## Outcomes

Our primary outcome was the grade of acute kidney injury (AKI) 48 postoperative hours after surgery [28]. Our secondary outcomes were the grade of AKI at 7 days and need for

postoperative RRT at any time after surgery. We also included time to first extubation, time to ICU discharge and survival up to 1 year as other secondary outcomes.

## Covariables

We collected patients' demographic characteristics, liver disease diagnosis and comorbidities as baseline characteristics. We included many perioperative variables we considered potential confounders that might influence both hemodynamic management interventions performed by anesthesiologists and our outcomes outside of the causal pathway. We included: age, sex, diabetes, severity of liver failure (Model for End Stage Liver Disease Sodium (MELD)), acute liver failure as a transplantation indication, retransplantation status, preoperative hemoglobin concentration, preoperative creatinine concentration, baseline intraoperative central venous pressure (CVP), the use of an intraoperative phlebotomy, length and type of vena cava clamping, length of graft cold ischemia time (CIT) and intraoperative exposure to starch (see S2 Appendix).

## Data sources and measurement

We prospectively collected age, sex, intraoperative volume of fluid received, intraoperative bleeding, diuresis, type of fluid used (crystalloids, synthetic colloids, non-synthetic colloids, red blood cells and other blood products), the use of a phlebotomy, preoperative creatinine and hemoglobin levels using a standardized case report form into a transplantation registry [21, 29–31]. We completed the registry with retrospective chart review for preoperative severity of liver failure (MELD), ascites, clamping type and length, cold ischemia time and all our outcomes, as reported in our previous publication [14]. For this study, we retrospectively collected vasopressor infusion doses at ICU admission from charts of included patients in the registry. Extracted data was kept in a coded dataset in an institutional secured server.

## Sample size

We used the available sample of patients gathered for our previous publication to observe an odds ratio of 1.05 or less in favor of a more restrictive fluid balance with an estimated incidence proportion of 48-hour AKI of 30% (with a power of 80%, a two-sided alpha level of 5% and the inclusion of 11 covariates) [14]. The available cohort was of 532 consecutive patients and all patients were included in our analyses.

## Data analyses

We reported categorical variables as proportions and continuous variables as means with standard deviations or medians with interquartile range based on skewness. We compared fluid balance between groups with and without vasopressors by a bivariate linear regression model. Our outcome variables were analyzed as either ordinal (AKI), dichotomic (RRT) or as time-to-event (extubation in the ICU, ICU stay and survival). For our primary analysis, we fitted a multivariable proportional odds regression model with the category of AKI (0, 1, 2, 3) as the dependent variable. A similar model was fitted for our secondary AKI outcome (7-day AKI). We fitted a logistic regression model for our RRT outcome. We used Fine and Gray models for time to first extubation and time to ICU discharge outcomes because they are competing risks with death and a Cox model was used for 1-year survival. We fitted all models by including vasopressor use (yes or no), fluid balance and considered their interaction by including the product term between the two as independent variables, as well as the aforementioned confounders. When statistical interaction (effect modification) was statistically significant, we

fitted stratified models presenting the fluid balance effect by vasopressors subgroups while adjusting the resulting models for the vasopressor dose as a continuous variable. As sensitivity analyses, we fitted all models without significant statistical interaction with the vasopressor dose as a continuous variable in equivalent of 10 μg/min of norepinephrine and fitted all models on complete cases only. We also explored confounding between vasopressor and fluid balance by a change-in-estimate approach.

Because some patients had repeated transplantations and were included more than once in the cohort, our analyses were fitted by Generalized Estimating Equations (GEE) (AKI and RRT) using robust standard errors with an exchangeable correlation matrix or by time-to-event models (time to extubation, time to ICU discharge and survival) using Sandwich robust standard errors to consider within-patient correlation. We evaluated the odds proportionality assumption for vasopressors in our ordinal models by a likelihood ratio test and the proportionality of risks assumption in our time-to-event models by the Harrell and Lee test and a visual inspection of the Schoenfeld residuals. Because this assumption was not always met, we used confounder stratification, coefficient-time interaction or a phlebotomy propensity score (that removed some variables from the model) to fit valid models. We used a phlebotomy propensity score since this variable was considered as a component of the intraoperative fluid management strategy. We explored non-linear effects of both fluid balance and vasopressors for all models. Multicollinearity was evaluated with the variance inflation factor. To handle missing values, we fitted all our models on 5 datasets with imputations by chained equations and pooled our estimates and standard errors using the Rubin's rule. To compare baseline characteristics between patients with and without vasopressors, we also conducted exploratory student T tests, Mann-Whitney U tests or chi-square tests and reported them in our supplementary material. Our alpha level was set at 0.05 and we reported all results with 95% confidence intervals. We used R (R Core Team, version 3.6.1) for all statistical analyses.

## Results

A total of 532 liver transplantations were available in our registry after exclusions. 527 transplantations were alive at ICU admission and 524 transplantations performed in 486 patients had data on vasopressors (S2 Fig in S3 Appendix). We summarized patients' characteristics in S1 and S2 Tables in S3 Appendix for supplementary figures and tables. Two hundred and forty-one transplantations (46%) had at least one vasopressor infusion at ICU admission. Among them, 140 (58%) had a vasopressin infusion and 228 had a norepinephrine infusion (95%). The median [IQR] norepinephrine equivalent dose was 25 [10, 38] μg per minute. Patients with vasopressors at the end of surgery were older, had a higher baseline MELD, a lower hemoglobin concentration and received less often a phlebotomy (Table 1). They also had a longer surgery with more blood loss, higher volumes of infused fluid, a higher volume of drained ascites and more red blood cell transfusions (Table 1). However, patients with vasopressors at the end of surgery had a lower fluid balance (mean (SD) of 0.2 L (3.9)) compared to patients without vasopressors (mean (SD) of 1.1 L (3.3)) (p = 0.015, Fig 1).

We provided descriptive outcome results in Table 2 and results from our models in Tables 3–5 and S3 and S4 Tables in S3 Appendix. For our primary outcome, neither the presence nor the dose of vasopressors at the end of surgery was associated with 48-hour AKI (Table 3). Vasopressor did not modify the effect of fluid balance and fluid balance was not associated with 48-hour AKI.

For our secondary outcomes, neither vasopressors at the end of surgery nor fluid balance was associated with 7-day AKI (Table 3). Vasopressor did not modify the effect of fluid balance on 7-day AKI. However, vasopressors use at the end of surgery significantly modified the fluid

**Table 1.  Important perioperative variables by vasopressor groups.**

| | No vasopressors (n = 283) | Vasopressors (n = 241) |
|---|---|---|
| **Baseline characteristics** | | |
| Age | 51 (12) | 54 (11) |
| Sex (male) | 195 (69%) | 158 (66%) |
| Hemoglobin level (g/L) | 109 (25) | 102 (23) |
| Bilirubin level (umol/L) | 53 [25, 157] | 81 [40, 166] |
| Creatinine level (umol/L) | 74 [59, 98] | 85 [66, 115] |
| INR | 1.5 [1.2, 1.9] | 1.6 [1.3, 2.1] |
| Sodium (mmol/L) | 136 (5) | 135 (6) |
| MELD | 19 (9) | 22 (8) |
| Acute liver failure | 15 (5%) | 8 (3%) |
| Retransplantation | 31 (11%) | 24 (10%) |
| **Donor characteristics** | | |
| Age | 51 (17) | 53 (16) |
| Sex (male)[+] | 158 (56%) | 133 (55%) |
| CIT (hours) | 7.4 (2.2) | 7.4 (2.5) |
| Type of donation: | | |
| • Donation after NDD | 280 (99%) | 233 (97%) |
| • DCD | 0 (0%) | 2 (1%) |
| • Living | 3 (16%) | 6 (2%) |
| Cause of death (excludes living donation) | | |
| • Anoxia | 46 (16%) | 46 (19%) |
| • Hemorrhagic stroke | 40 (14%) | 38 (16%) |
| • Ischemic stroke | 97 (34%) | 87 (36%) |
| • Subarachnoid hemorrhage | 18 (6%) | 17 (7%) |
| • Traumatic brain injury | 68 (24%) | 40 (17%) |
| • Other | 7 (2%) | 4 (2%) |
| • Not reported cause of NDD | 4 (1%) | 3 (1%) |
| **Surgical variables** | | |
| Vena cava clamping time (minutes) | 40 (17) | 43 (19) |
| Length of surgery (minutes) | 230 [200, 266] | 240 [210, 300] |
| Piggyback caval anastomosis | 9 (3%) | 14 (6%) |
| **Anesthesiologic variables** | | |
| Baseline CVP (mmHg) | 14.2 (5.4) | 14.5 (5.0) |
| Phlebotomy (exposed) | 179 (63%) | 115 (48%) |
| Ascites (L)[++] | 0.5 [0, 4.0] | 2.0 [0.5, 6.0] |
| Intraoperative urine output (L) | 0.35 [0.22, 0.54] | 0.35 [0.20, 0.50] |
| Intraoperative bleeding (L) | 0.8 [0.5, 1.3] | 1.4 [0.8, 2.5] |
| Crystalloid (L) | 3.8 [3.0, 4.5] | 4.0 [3.0, 5.0] |
| Colloid (L) | 0.5 [0, 0.5] | 0.5 [0, 1.0] |
| Cellsaver output (L) | 0.20 [0.10, 0.35] | 0.32 [0.18, 0.51] |
| Intraoperative RBC transfusions (%) | 45 (16%) | 94 (39%) |
| **Main exposure** | | |
| Fluid balance (L)[+++] | 1.1 (3.3)[*] | 0.2 (3.9)[*] |
| Norepinephrine infusion upon ICU admission | | 228 (95%) |
| Vasopressin infusion upon ICU admission | | 140 (58%) |

(*Continued*)

**Table 1.** (Continued)

| | No vasopressors (n = 283) | Vasopressors (n = 241) |
|---|---|---|
| Any inotrope upon ICU admission | | 3 (1%) |

Results are reported as number of observed cases (proportion in %), as means (SD) or as medians [quartile 1, quartile 3].

N.B. 8 missing values for vasopressors. Missing values < 5 per group are not reported.

[+] 5 (2%) and 2 (1%) missing values respectively.

[++] 55 (19.4%) and 42 (17.4%) missing values respectively.

[+++] 58 (20.5%) and 44 (18.3%) missing values respectively.

[*] Means difference for fluid balance had a p value = 0.015 when fitted in a GEE linear model. See S2 Table in S3 Appendix for statistical tests.

Abbreviations: INR = International Normalized ratio, MELD = Model for End-stage Liver Disease, CIT = Cold Ischemia Time, NDD = Neurological Determination of Death, DCD = Donation after Circulatory Death, CVP = Centre Venous Pressure, RBC = Red Blood Cells, ICU = intensive care unit.

balance effect on two of our secondary outcomes: risk of postoperative RRT and 1-year survival (Tables 4 and 5). In patients with vasopressors, fluid balance and dose of vasopressors were not associated with the risk of postoperative RRT, while in the subgroup of patients without vasopressors, a higher fluid balance was significantly associated with a higher risk of postoperative RRT (OR = 1.52 [1.15, 2.01]), p for interaction < 0.001, (Table 4)). For our 1-year survival outcome, a higher fluid balance was significantly associated with a higher mortality (HR = 1.71 [1.26, 2.34]) in patients without vasopressors. However, in patients with vasopressors, a higher fluid balance was not associated with mortality (HR = 1.24 [0.99, 1.53]) while higher doses of vasopressors were (HR = 1.29 [1.13, 1.47] per 10 µg/min of norepinephrine) (Table 5 and S3 Fig in S3 Appendix). Fluid balance had thus a lesser effect on mortality in the subgroup of patients with vasopressors (p < 0.01 for interaction) (Table 5).

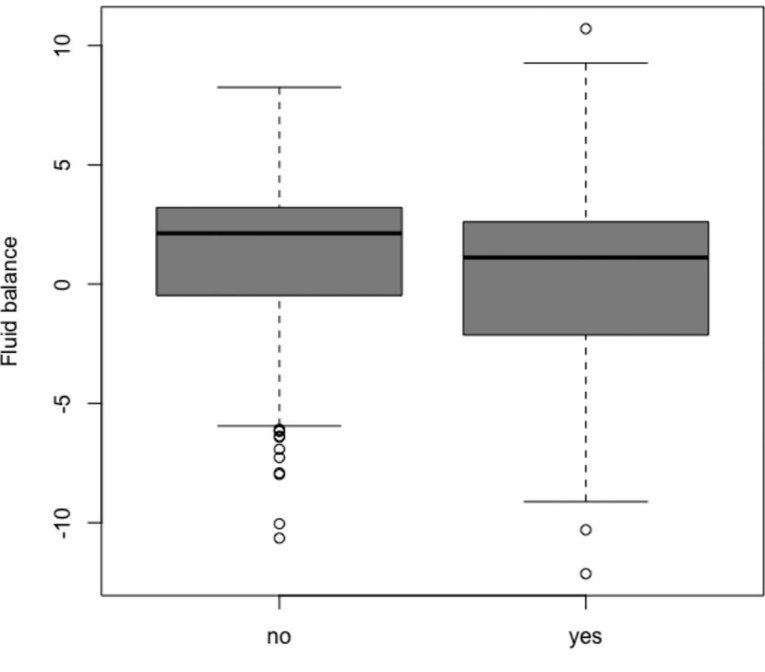

**Fig 1. Fluid balance according to the presence of vasopressors.** Boxplots report medians and interquartile ranges. The observed difference between groups has a p value = 0.015 when fitted in a GEE linear model. Detailed results regarding fluid are summarized in Table 1.

**Table 2. Postoperative outcomes by vasopressor groups used in multivariable analyses.**

|  | No vasopressors (n = 283) | Vasopressors (n = 241) |
|---|---|---|
| Need for postoperative RRT | | |
| Postoperative RRT | 12 (4.2%) | 18 (7.5%) |
| AKI results– 48 hours* | | |
| 0 | 92 (32.6%) | 63 (26.4%) |
| 1 | 32 (11.4%) | 32 (13.4%) |
| 2 | 80 (28.4%) | 73 (30.5%) |
| 3 | 78 (27.7%) | 71 (29.7%) |
| AKI results—7 days** | | |
| 0 | 193 (69.2%) | 154 (65.3%) |
| 1 | 23 (8.2%) | 17 (7.2%) |
| 2 | 40 (14.3%) | 31 (13.1%) |
| 3 | 23 (8.2%) | 34 (14.4%) |
| Creatinine values (umol/L) | | |
| 48 hours* | 102 [69, 159] | 125 [84, 189] |
| 7 days** | 78 [59, 101] | 80 [60, 104] |
| Other postoperative data*** | | |
| Time to first extubation (hours)[a] | 6.6 [5.8, 7.6] | 10.8 [8.5, 12.8] |
| Time to ICU discharge (days)[b] | 2.8 [2.6, 3.3] | 3.2 [2.9, 3.6] |
| Time to hospital discharge (days)[c] | 27.8 [26.5, 30.3] | 30.8 [28.2, 34.6] |
| Survival | | |
| Survival up to 1 year[d] | 253 (93.7%) | 212 (92.2%) |

Results are presented as proportions or as medians with [q1, q3].

AKI results are based on either the creatinine change criteria or the urine output. For observations with missing data on urine output, AKI is classified only based on the creatinine change criteria.

* 3 missing values because of death before 36 hours

** 14 missing values because of death within 7 days

*** Results are reported as Kaplan-Meir medians with 95% confidence limits (with death considered as censor)

[a] n = 526 transplantations

[b] n = 530 transplantations

[c] n = 531 transplantations

[d] n = 486 patients who received 524 transplantations (270 non duplicated patients who did not have vasopressors and 230 non duplicated patients who had vasopressors (some patients had a transplantation in each group))

Abbreviations: RRT = renal replacement therapy, AKI = acute kidney injury, ICU = intensive care unit

Vasopressors at the end of surgery use did not modify the effect of fluid balance on our time to extubation and time to ICU discharge outcomes. The presence of any vasopressor at the end of surgery was associated with a lower risk of being extubated in the ICU (HR = 0.64 [0.51, 0.81]) while fluid balance did not have any effect (S3 Table in S3 Appendix). Vasopressors use were not associated with a lower risk of being discharge from ICU (HR = 0.91 [0.76, 1.09]), but a higher fluid balance was associated with a lower risk of being discharge from ICU (non-linear effect, p < 0.001) (S4 Table and S4 Fig in S3 Appendix).

In our sensitivity analyses, results were robust for most outcomes (S5 and S6 Tables in S3 Appendix). For the time to ICU discharge, vasopressors had an effect when analyzed as a continuous variable (HR = 0.92 [0.87, 0.97] per 10 μg/min of norepinephrine) (S5 Table in S3 Appendix). For our time to extubation outcome, the presence of vasopressor had no more a

**Table 3. AKI.**

| Variables | 48-hour AKI (POR) (n = 527) | 7-day AKI (POR) (n = 521) |
|---|---|---|
| Any vasopressors upon ICU admission | 1.01 [0.72, 1.40] | 1.10 [0.76, 1.61] |
| Fluid balance (L) | 0.99 [0.94, 1.04] | 1.04 [0.97, 1.10] |
| Intraoperative phlebotomy | 0.87 [0.60, 1.27] | 1.26 [0.80, 1.98] |
| Age (years) | 1.02 [1.01, 1.04]* | 1.02 [1.01, 1.04]* |
| Sex (male) | 0.88 [0.61, 1.26] | 1.00 [0.66, 1.52] |
| Retransplantation | 0.55 [0.30, 1.00] | 0.87 [0.44, 1.72] |
| ALF | 0.73 [0.30, 1.80] | 1.53 [0.67, 3.51] |
| MELD | 1.04 [1.02, 1.07]* | 1.03 [1.01, 1.06]* |
| Diabetes | 1.31 [0.87, 1.98] | 1.26 [0.78, 2.03] |
| Baseline CVP (mmHg) | 1.01 [0.98, 1.05] | 1.03 [0.99, 1.06] |
| CIT (hours) | 1.08 [1.01, 1.16]* | 1.07 [0.99, 1.17] |
| Vena cava clamping time (minutes) | 1.01 [0.99, 1.02] | 1.01 [0.99, 1.02] |
| Baseline hemoglobin (g/L) | 1.00 [0.99, 1.01] | 0.99 [0.98, 1.01] |
| Baseline creatinine (10 µmol/L) | 0.96 [0.92, 1.00] | 1.00 [0.96, 1.06] |
| Piggyback | 0.63 [0.28, 1.44] | 1.43 [0.62, 3.30] |
| Any intraoperative starch | 1.17 [0.85, 1.61] | 1.03 [0.71, 1.48] |

* Statistically significant at alpha = 0.05.

Results are expressed with 95% confidence intervals.

These results come from a multivariable adjusted for reported variables.

Interaction between fluid balance and presence of vasopressor was not significant.

Abbreviations: AKI = acute kidney injury, POR = proportional odds ratio, ALF = acute liver failure, MELD = Model for End-stage Liver Disease, CVP = central venous pressure, CIT = cold ischemia time

significant effect when analyzed on complete cases only (S7 Table in S3 Appendix). We reported change-in-estimate analyses to explore confounding effects (S7 Table in S3 Appendix) as well as the effect of interaction on survival (S8 Table in S3 Appendix).

**Table 4. Postoperative RRT.**

| Variables | OR (n = 527) |
|---|---|
| **Patients without vasopressor upon ICU admission** | |
| Fluid balance (L) | 1.52 [1.15, 2.01]* |
| Intraoperative phlebotomy | 0.61 [0.14, 2.67] |
| **Patients with vasopressors upon ICU admission** | |
| Vasopressor dose+ | 1.04 [0.91, 1.17] |
| Fluid balance (L) | 1.00 [0.86, 1.17] |
| Intraoperative phlebotomy | 0.26 [0.06, 1.19] |

+ In equivalent of 10 ug/min of norepinephrine.

* Statistically significant at alpha = 0.05.

Results are expressed with 95% confidence intervals.

p value for interaction < 0.001 between fluid balance and presence of any vasopressor at the end of surgery.

There were 283 patients with 12 RRT events in the subgroup without vasopressor and 244 patients with 18 RRT events in the subgroup with vasopressors. These results come from stratified multivariable models adjusted with a propensity score for phlebotomy (preoperative hemoglobin value, preoperative creatinine value, baseline CVP, age, preoperative MELD score and transplantation for acute liver failure).

Abbreviations: OR = odds ratio, ICU = intensive care unit.

**Table 5. 1-year survival.**

| Variables | HR (n = 527) |
|---|---|
| **Patients without vasopressor upon ICU admission** | |
| Fluid balance (L) | 1.71 [1.26, 2.34]* |
| Intraoperative phlebotomy | 0.28 [0.08, 0.98]* |
| **Patients with vasopressors upon ICU admission** | |
| Vasopressor dose[+] | 1.29 [1.13, 1.47]* |
| Fluid balance (L) | 1.24 [0.99, 1.53] |
| Vasopressor dose*time (month)[++] | 0.95 [0.90, 1.00] |
| Fluid balance*time (month)[++] | 0.97 [0.93, 0.99]* |

[+] In equivalent of 10 ug/min of norepinephrine.

[++] This suggests that the effect weans off over time.

* Statistically significant at alpha = 0.05

Results are expressed with 95% confidence intervals.

p values < 0.01 for interaction between fluid balance and presence of any vasopressor at the end of surgery.

There were 283 patients with 17 death events in the subgroup without vasopressor and 244 patients with 18 death events in the subgroup with vasopressors. These results come from stratified multivariable models. Both Cox models were fitted with a propensity score for exposure to a phlebotomy (preoperative hemoglobin value, preoperative creatinine value, baseline CVP, age, preoperative MELD score and transplantation for acute liver failure). Model B with vasopressor doses was also stratified for vasopressin infusion and exposure to a phlebotomy (for proportionality reasons).

## Discussion

This study evaluated the association between intraoperative hemodynamic management and postoperative outcomes in adult liver transplantations. Neither vasopressors nor fluid balance had any effect on AKI. The presence of vasopressors attenuated the effect of fluid balance on the risk of RRT and on survival. A higher fluid balance was associated with a higher risk of RRT only in patients without vasopressors. Both higher doses of vasopressors and a higher fluid balance were associated with a lower survival and the effect of fluid balance was worse in patients without vasopressors.

The observation that a higher fluid balance was associated with an increased risk of RRT and an increased risk of dying only in the subgroup of patients without vasopressors may suggest that either fluid balance had less effect when patients are hemodynamically unstable and received vasopressors or that a restrictive fluid management strategy that includes a liberal use of vasopressors might improve outcomes. Indeed, we observed a lower fluid balance in patients with vasopressors although they were sicker and had higher intraoperative blood loss. This observation might suggest that some of these patients were exposed to a more restrictive fluid management strategy that included a more liberal use of vasopressors. A similar association between fluid management strategies and need for vasopressors has already been suggested in other major surgeries [15, 20]. Because vasopressor use is so closely correlated to the use of fluid, further work is needed to better understand its effects in the context of a restrictive fluid management strategy.

Previous studies have suggested that vasopressors use were associated with worse outcomes after liver transplantation [32–34]. Other studies suggested that intraoperative hypotension and hemodynamic instability were also associated with worse postoperative outcomes [8–10]. In our study, patients who remained on vasopressors at the end of surgery had more blood loss and longer surgeries, suggesting that vasopressors are potentially markers of a more

difficult intraoperative clinical course with more intraoperative hemodynamic instability. The observed association between vasopressors and longer time before extubation, ICU discharge and worse survival is thus probably not causal. Even if vasopressors were markers of hemodynamic instability and severity of the underlying clinical course, their liberal use might have allowed a more restrictive fluid management strategy for the same hemodynamic goals as suggested by our observations. This hypothesis would be in line with similar observed effects in this population [13]. However, in this study, the effect of an intentional use of vasopressors to reduce intraoperative fluid balance may not be differentiated from the effect of vasopressors use in the context of hemodynamic instability.

The main strength of this study was the evaluation of the effects of a surrogate of the intraoperative hemodynamic management on the incidence of postoperative outcomes in liver transplantation; such a work had never been done in this population. We also used several strategies to limit biases by inclusion of consecutive patients and by limiting our exclusion criteria to patients for whom classification of the primary outcome was impossible (patients on RRT prior to transplantation). Classification of exposures and outcomes were made from a mix of prospectively collected and retrospectively extracted data from clinical charts, ensuring high-quality data. Finally, multiple sensitivity analyses were conducted to assess the robustness of our findings.

As for the limitations, we acknowledge a residual risk of non-differential classification bias if data was incorrectly entered in the clinical chart. Our fluid balance exposure is a composite exposure including the total volume of bleeding in the surgical suctions, which includes some ascites that is impossible to measure. This could have misclassified this variable by an artificial increase of the bleeding volume in the sicker patient with higher amount of ascites. Also, our fluid balance variable captured similar fluid balance in patients with different blood loss and could have induced non-differential classification bias in any direction. Many residual potential confounders might influence the association between intraoperative fluid balance or vasopressor doses and our outcomes. For example, we did not capture vasopressor doses at different periods of the surgery, such as during the dissection phase when a phlebotomy is used in around 50% of the patients, acute hemodynamics changes and their severity, pre-emptive actions based on surgical maneuvers and the clinical feeling of anesthesiologists, all important decision-making variables that might affect fluid administration and intraoperative vasopressor use. Vasopressors were also used to treat hemodynamic instability and we did not include any intraoperative hemodynamic variable in our models. We however adjusted our statistical models for many selected covariables potentially associated with both exposure and outcome outside the causal pathway. This study was conducted in only one center, which limits its generalizability to other populations. Finally, we conducted multiple statistical testing that could have produced significant results by chance alone. Thus, only robust and concordant results should be interpreted accordingly.

In conclusion, this study provides new information regarding the effects of intraoperative fluid and vasopressor administration on the postoperative course of liver transplant recipients. The liberal use of vasopressors to implement a restrictive fluid management strategy is a potential therapeutic avenue for these patients and deserves further investigation.

## Supporting information

**S1 Appendix. Donor consent form in the Province of Quebec.**
(PDF)

**S2 Appendix. Intraoperative management, selection of confounders and outcomes and conceptual framework of the research question.**
(DOCX)

**S3 Appendix. Supplementary tables and figures.**
(DOCX)

# Acknowledgments

We would like to thank all anesthesiologists from the CHUM for the quality of data they gathered over time.

# Author Contributions

**Conceptualization:** François Martin Carrier, Marc Bilodeau, Michaël Chassé.

**Data curation:** François Martin Carrier, Luc Massicotte.

**Formal analysis:** François Martin Carrier, Marie-Pierre Sylvestre, Michaël Chassé.

**Funding acquisition:** François Martin Carrier.

**Investigation:** François Martin Carrier.

**Methodology:** François Martin Carrier, Michaël Chassé.

**Supervision:** Marc Bilodeau, Michaël Chassé.

**Writing – original draft:** François Martin Carrier, Marie-Pierre Sylvestre, Luc Massicotte, Marc Bilodeau, Michaël Chassé.

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
