## [Decision Letter · Decision Letter 0]

1 Jul 2020

PONE-D-20-12508

Effects of intraoperative hemodynamic management on postoperative acute kidney injury in liver transplantation: an observational cohort study.

PLOS ONE

Dear Dr. Carrier,

Thank you for submitting your manuscript to PLOS ONE. After careful consideration, we feel that it has merit but does not fully meet PLOS ONE’s publication criteria as it currently stands. Therefore, we invite you to submit a revised version of the manuscript that addresses the points raised during the review process

We look forward to receiving your revised manuscript.

Kind regards,

Richard Hodge (on behalf of Academic Editor Ehab Farag, MD FRCA FASA)

Associate Editor

PLOS ONE

Journal Requirements:

2. In your ethics statement, please provide additional information about the patient records used in your retrospective study.

Specifically, please ensure that you have discussed whether all data were fully anonymized before you accessed them.

3. We note that your study involved tissue/organ transplantation. Please provide the following information regarding tissue/organ donors for transplantation cases analyzed in your study.  

a. Please provide the source(s) of the transplanted tissue/organs used in the study, including the institution name and a non-identifying description of the donor(s).

b. Please state in your response letter and ethics statement whether the transplant cases for this study involved any vulnerable populations; for example, tissue/organs from prisoners, subjects with reduced mental capacity due to illness or age, or minors.

- If a vulnerable population was used, please describe the population, justify the decision to use tissue/organ donations from this group, and clearly describe what measures were taken in the informed consent procedure to assure protection of the vulnerable group and avoid coercion.  

- If a vulnerable population was not used, please state in your ethics statement, “None of the transplant donors was from a vulnerable population and all donors or next of kin provided written informed consent that was freely given.”

4. In the Methods, please provide detailed information about the procedure by which informed consent was obtained from organ/tissue donors or their next of kin. In addition, please provide a blank example of the form used to obtain consent from donors, and an English translation if the original is in a different language.

5. Please indicate whether the donors were previously registered as organ donors. If tissues/organs were obtained from deceased donors or cadavers, please provide details as to the donors’ cause(s) of death.

6. Please provide the participant recruitment dates and the period during which transplant procedures were done (as month and year).

7. Please discuss whether medical costs were covered or other cash payments were provided to the family of the donor. If so, please specify the value of this support (in local currency and equivalent to U.S. dollars).

8. We noted in your submission details that a portion of your manuscript may have been presented or published elsewhere:

'Results presented in this manuscript have not been published elsewhere. However, analyses from the same cohort of patients have been previously published (Carrier FM et al. Effects of vasopressors on postoperative acute kidney injury after liver transplantation: an observational cohort study. Transplantation 2019).'

Please clarify whether this  publication was peer-reviewed and formally published.

If this work was previously peer-reviewed and published, in the cover letter please provide the reason that this work does not constitute dual publication and should be included in the current manuscript.

9. We note that you have indicated that data from this study are available upon request. PLOS only allows data to be available upon request if there are legal or ethical restrictions on sharing data publicly. For information on unacceptable data access restrictions, please see http://journals.plos.org/plosone/s/data-availability#loc-unacceptable-data-access-restrictions.

10. Please amend either the title on the online submission form (via Edit Submission) or the title in the manuscript so that they are identical.

11. Please include captions for your Supporting Information files at the end of your manuscript, and update any in-text citations to match accordingly. Please see our Supporting Information guidelines for more information: http://journals.plos.org/plosone/s/supporting-information

Reviewers' comments:

Reviewer's Responses to Questions

**Comments to the Author**

1. Is the manuscript technically sound, and do the data support the conclusions?

Reviewer #1: Yes

2. Has the statistical analysis been performed appropriately and rigorously? 

Reviewer #1: Yes

3. Have the authors made all data underlying the findings in their manuscript fully available?

Reviewer #1: Yes

4. Is the manuscript presented in an intelligible fashion and written in standard English?

Reviewer #1: Yes

5. Review Comments to the Author

Reviewer #1: Excellent paper on the subject of AKI after orthotopic liver transplantation. the authors compared outcome of vasopressors with volume resuscitation during liver transplant.

6. PLOS authors have the option to publish the peer review history of their article (what does this mean?). If published, this will include your full peer review and any attached files.

Reviewer #1: No

---

## [Author Response · Author response to Decision Letter 0]

14 Jul 2020

Dear Editorial Board,

Thank you for the thorough revision of our manuscript. You may find below detailed answers to comments. As requested, some of the comments were addressed directly in the cover letter.

We hope we addressed all raised concerns.

Best regards,

François M. Carrier and co-authors

Editor’s comments

We revised the manuscript accordingly. 

2. In your ethics statement, please provide additional information about the patient records used in your retrospective study. Specifically, please ensure that you have discussed whether all data were fully anonymized before you accessed them.

We had access to patients’ chart numbers to be able extract needed data from patients electronic medical file. Some data was available in a data set already authorized by our IRB and other data had to be manually extracted from electronic file. We did not keep any nominal information outside chart numbers. At the end of the work, we created a coded data set (patient ID number with corresponding chart number in a different protected file) that we kept in a secured server at the research center. 

We added more information regarding this in the “data sources and measurement” section in page 10.

3. We note that your study involved tissue/organ transplantation. Please provide the following information regarding tissue/organ donors for transplantation cases analyzed in your study. 

a. Please provide the source(s) of the transplanted tissue/organs used in the study, including the institution name and a non-identifying description of the donor(s).

Organ donation in the Province of Quebec is managed by our national organ procurement organization, Transplant Québec. Most organs are allocated within the province, but some of them may come from other Canadian provinces. Donors information is completely anonymized and unknow to recipients. 

However, for research purposes, some anonymized donor characteristics were available in our institutional database. This data was not available at the time the first version of this manuscript was written but is now available through our institutional data management system. After authorization from our IRB, we accessed this data and extracted available donor characteristics. We added them in table 1 in page 14 of the manuscript (as well as in table S1) to describe the donor population. 

b. Please state in your response letter and ethics statement whether the transplant cases for this study involved any vulnerable populations; for example, tissue/organs from prisoners, subjects with reduced mental capacity due to illness or age, or minors.

- If a vulnerable population was used, please describe the population, justify the decision to use tissue/organ donations from this group, and clearly describe what measures were taken in the informed consent procedure to assure protection of the vulnerable group and avoid coercion. 

- If a vulnerable population was not used, please state in your ethics statement, “None of the transplant donors was from a vulnerable population and all donors or next of kin provided written informed consent that was freely given.”

We added this sentence in the first paragraph of the methods section (page 8). 

4. In the Methods, please provide detailed information about the procedure by which informed consent was obtained from organ/tissue donors or their next of kin. In addition, please provide a blank example of the form used to obtain consent from donors, and an English translation if the original is in a different language.

We added a sentence about this (page 8) and provided a blank example of the form used to obtain consent in the supplementary material (appendix 1). 

5. Please indicate whether the donors were previously registered as organ donors. If tissues/organs were obtained from deceased donors or cadavers, please provide details as to the donors’ cause(s) of death.

Some donors were previously registered in either the Régie de l’assurance maladie du Québec (RAMQ) organ donor registry or in the organ donor registry managed by the Chambre des notaires du Québec (https://www.transplantquebec.ca/node/107). For those who were not registered, consent was obtained from their next of kin who may act as the surrogate decision maker according to the Quebec medical law. For those who were registered, consent was also obtained from the surrogate decision maker; in Quebec, registries are only informative of patient willingness. 

In recipients’ charts, we do not have the information regarding donor registration status. Donor information is strictly confidential and only medical information regarding donor characteristics were available and cross-linked as previously described. 

We added a sentence in page 8 to better explain this. 

6. Please provide the participant recruitment dates and the period during which transplant procedures were done (as month and year).

The period during which transplant procedures were done was already reported in the “setting and participant sub-section” of the “methods” section. We added dates of data extraction in page 7.

7. Please discuss whether medical costs were covered or other cash payments were provided to the family of the donor. If so, please specify the value of this support (in local currency and equivalent to U.S. dollars).

In Canada, no financial reimbursement is provided to donor families. Our health system is 100% public and nationalized. We added a sentence in page 8 to report this.

8. We noted in your submission details that a portion of your manuscript may have been presented or published elsewhere:

'Results presented in this manuscript have not been published elsewhere. However, analyses from the same cohort of patients have been previously published (Carrier FM et al. Effects of vasopressors on postoperative acute kidney injury after liver transplantation: an observational cohort study. Transplantation 2019).'

Please clarify whether this publication was peer-reviewed and formally published.

Yes, this publication was peer-reviewed and formally published. We updated the reference (Carrier FM, Chassé M, Sylvestre M-P, Girard M, Legendre-Courville L, Massicotte L, et al. Effects of Intraoperative Fluid Balance During Liver Transplantation on Postoperative Acute Kidney Injury: An Observational Cohort Study. Transplantation. 2020;104: 1419–1428.) since the previous one was a “online first” publication. 

If this work was previously peer-reviewed and published, in the cover letter please provide the reason that this work does not constitute dual publication and should be included in the current manuscript.

Done. Please see the cover letter. All reported results in this manuscript are genuine and new and address a new research question. 

9. We note that you have indicated that data from this study are available upon request. PLOS only allows data to be available upon request if there are legal or ethical restrictions on sharing data publicly. For information on unacceptable data access restrictions, please see http://journals.plos.org/plosone/s/data-availability#loc-unacceptable-data-access-restrictions.

Done. Data may be available upon request only, as per national regulations (province of Quebec). Please see the cover letter. 

10. Please amend either the title on the online submission form (via Edit Submission) or the title in the manuscript so that they are identical.

Done.

11. Please include captions for your Supporting Information files at the end of your manuscript, and update any in-text citations to match accordingly. Please see our Supporting Information guidelines for more information: http://journals.plos.org/plosone/s/supporting-information

Done.

---

## [Editor Report · Decision Letter 1]

29 Jul 2020

Effects of intraoperative hemodynamic management on postoperative acute kidney injury in liver transplantation: an observational cohort study

PONE-D-20-12508R1

Dear Dr. François Martin Carrier

We’re pleased to inform you that your manuscript has been judged scientifically suitable for publication and will be formally accepted for publication once it meets all outstanding technical requirements.

Kind regards,

Ehab Farag, MD FRCA FASA

Academic Editor

PLOS ONE
---

## [Editor Report · Acceptance letter]

30 Jul 2020

PONE-D-20-12508R1 

Effects of intraoperative hemodynamic management on postoperative acute kidney injury in liver transplantation: an observational cohort study 

Dear Dr. Carrier:

I'm pleased to inform you that your manuscript has been deemed suitable for publication in PLOS ONE. Congratulations! Your manuscript is now with our production department. 

Kind regards, 

on behalf of

Dr. Ehab Farag 

Academic Editor

PLOS ONE